# System-level clustering of testosterone-related biomarkers identifies high-risk aging profiles linked to inflammation and renal function
Nobuo Okui [1,2,3] ✉ & Shigeo Horie[1,3,4]

## Abstract

**Background** Serum total testosterone (TT) interacts with multiple physiological systems and is implicated in heterogeneous aging processes in men. However, aging-related phenotypes associated with TT are unlikely to be captured by single biomarkers or conventional clinical categories. This study aims to identify data-driven aging phenotypes based on TT and related clinical biomarkers using an unsupervised analytical framework.

**Methods** Clinical laboratory data from 5,877 Japanese male patients undergoing routine health evaluations are analyzed. After restricting the cohort to individuals with complete age and body mass index data, missing values in other variables are imputed using column-wise mean imputation. Unsupervised clustering is performed using K-means on standardized biomarkers related to endocrine, metabolic, inflammatory, and renal function. Principal component analysis and correlation network analysis are used for visualization. External validation uses cancer prevalence data from the NHANES dataset.

**Results** Four physiological clusters are identified. One cluster shows low TT levels, elevated inflammatory markers, impaired renal function, and higher cancer prevalence in external validation, indicating a high-risk aging profile. Other clusters show preserved hormonal and metabolic profiles. Network analysis reveals cluster-specific differences in the centrality of TT within physiological networks. Principal component analysis shows overlapping cluster distributions, reflecting continuous aging-related variation.

**Conclusions** Unsupervised clustering of TT-related biomarkers reveals aging phenotypes beyond conventional clinical classifications. TT functions as part of an integrated physiological network rather than as an isolated marker. These findings support a systems-level perspective on male aging and demonstrate utility of data-driven phenotyping, while acknowledging the descriptive and cross-sectional nature of the analysis.

## Plain language summary

Testosterone is commonly measured in adult men, but its clinical meaning is often unclear when considered in isolation. Similar testosterone levels can reflect very different physiological states depending on inflammation, kidney function, and other biological factors. In this study, routine blood test data from more than 5,800 Japanese men were analyzed to examine how testosterone relates to other clinical biomarkers. Using an unsupervised machine learning approach, individuals were grouped based on combined patterns of testosterone, inflammatory markers, renal function, and metabolic measures, rather than on testosterone levels alone. Several distinct physiological contexts of testosterone were identified. In one group, low testosterone was accompanied by elevated inflammation and impaired kidney function, suggesting a coordinated biological state rather than an isolated hormone abnormality. When similar biomarker profiles were examined in an independent U.S. population dataset, individuals with this profile showed higher cancer prevalence, supporting consistency across populations. These findings indicate that measuring testosterone is most informative when interpreted alongside other routinely available biomarkers, rather than as a standalone test.

[1]Innovative Longevity, Juntendo University, Bunkyo, Tokyo, Japan. [2]Data Science, Kanagawa Dental University, Yokosuka, Kanagawa, Japan. [3]Data Science and Informatics for Genetic Disorders, Juntendo University, Bunkyo, Tokyo, Japan. [4]Urology, Juntendo University, Bunkyo, Tokyo, Japan. ✉e-mail: okuinobuo@gmail.com

Aging in men is accompanied by heterogeneous physiological changes across endocrine, metabolic, inflammatory, and renal systems[1–4]. Among these, serum total testosterone (TT) has attracted sustained clinical interest because of its associations with muscle mass, metabolic regulation, cardiovascular health, bone density, and hematological function[5–8]. Although age-related decline in TT is well documented, its clinical significance varies substantially among individuals[3,9,10]. Some men maintain relatively stable physiological function despite lower TT levels, whereas others exhibit concurrent abnormalities across multiple biological domains[11]. These observations indicate that aging-related physiological change cannot be adequately captured by chronological age or by any single biomarker alone, underscoring the heterogeneity of biological aging processes[3,10–12].

Most existing studies have examined TT within predefined disease categories or through univariate or regression-based analytical frameworks[11,13]. While such approaches have clarified specific associations —for example, between TT and metabolic or inflammatory markers[14–16]— they may overlook coordinated, system-level patterns that emerge across physiological domains during aging[4,17,18]. Aging is increasingly understood as a multivariate process involving interactions among endocrine, metabolic, inflammatory, and renal systems rather than isolated organ-specific changes[2,19,20]. Accordingly, data-driven approaches that enable identification of latent physiological patterns without imposing predefined disease labels may offer complementary insights into aging-related variation[21–24].

An important unresolved question is whether such multivariate physiological patterns correspond to clinically meaningful differences[25–28]. Identification of biomarker combinations is of limited value unless these patterns demonstrate consistency across populations or relate to relevant health outcomes[29–32]. In exploratory studies aimed at characterizing heterogeneity rather than inferring causality, external validation using robust and well-documented clinical outcomes provides a pragmatic strategy for assessing real-world relevance[33–35]. Cancer was selected in this context because it represents a clinically established, system-level outcome reflecting long-term effects of inflammation, endocrine dysregulation, and organ function, and because cancer history is consistently available in large population-based datasets[35–39].

In this study, an exploratory, multi-stage analytical framework was applied to characterize TT-centered physiological patterns, given that TT and related clinical contexts are linked to clinically relevant heterogeneity in cardiometabolic risk, inflammatory status, and overall health in aging men[40–42]. First, unsupervised clustering was used to identify subgroups based on multivariate biomarker profiles without prespecifying disease categories[43–47]. Second, cluster-specific relationships were examined using correlation-based network analysis to describe differences in physiological structure, emphasizing descriptive patterns rather than causal inference[48–52]. Third, external consistency of selected biomarker-defined profiles was assessed using cancer prevalence in an independent population-based dataset (NHANES), to evaluate whether these multivariate patterns capture clinically relevant heterogeneity beyond chronological age alone[37,39,53].

## Methods
### Ethical approval
This cross-sectional study was approved by the Hospital Research Ethics Board of Juntendo University, Faculty of Medicine (H19-0128) and conducted in accordance with the Declaration of Helsinki and applicable Japanese ethical guidelines. This study was approved by the institutional ethics committee, and informed consent was waived because the analysis was based on retrospective, anonymized clinical data with no direct patient contact.

### Step 1
**Unsupervised clustering of multivariate biomarker profiles.** Data were obtained from outpatients who underwent TT measurement at Juntendo University Hospital (Tokyo, Japan) between 2008 and 2019 ($N = 12,547$). Female patients and males younger than 20 years were excluded. Inclusion required complete age and body mass index (BMI)

data; no exclusions were applied for comorbidities, medications, or diagnoses. The final cohort comprised 5,854 male patients aged 20–98 years. For each individual, one record was selected corresponding to the date with the largest number of laboratory tests; if multiple dates met this criterion, the earliest date was used. Laboratory measurements were obtained under routine clinical protocols (Supplementary Table S1).

**Handling of missing data.** Age and BMI were complete by design; however, missingness remained in other laboratory variables, with variable-specific missing rates ranging from 2.9 to 73.1% (Supplementary Table S2). To avoid case-wise deletion and preserve cohort size, missing laboratory values were imputed using column-wise mean imputation before multivariate analyses. All downstream procedures were conducted on the same imputed and standardized dataset to ensure internal consistency; these included hierarchical clustering, K-means clustering, and principal component analysis (PCA) for visualization. Robustness to imputation strategy was assessed using k-nearest neighbors (KNN) imputation and iterative imputation; overall clustering patterns were broadly preserved across methods, and column-wise mean imputation was retained for the primary analyses.

**Exploratory hierarchical clustering.** Hierarchical clustering (Ward linkage; Euclidean distance) was applied to the standardized multivariate dataset as an exploratory visualization of global similarity structure. This analysis was used for qualitative inspection only and was not used to define cluster membership or select the number of clusters.

**K-means clustering and justification for selecting $K = 4$.** Formal cluster assignment was performed using K-means clustering in the full standardized feature space. The number of clusters was selected by integrating multiple criteria: within-cluster sum of squares (elbow), silhouette coefficients, stability across random initializations, and clinical separability of biomarker-defined patterns. The elbow method showed a clear inflection around $K = 4$. Although the average silhouette coefficient was marginally higher at $K = 3$ than at $K = 4$, the absolute difference was small. Importantly, centroid comparisons demonstrated that the $K = 4$ solution did not simply over-fragment the data: individuals characterized by low TT and elevated C-reactive protein (CRP) formed a single group under $K = 3$, whereas $K = 4$ consistently separated this group into two subgroups that shared low TT and high CRP but differed in renal function (serum creatinine). Because the study objective was to characterize clinically interpretable physiological heterogeneity rather than to optimize a single clustering metric, $K = 4$ was adopted as the primary solution. A singleton cluster was retained as an exploratory subgroup but excluded from inferential statistical comparisons.

**PCA for visualization.** PCA was applied solely for visualization to project individuals onto the first two principal components and to facilitate graphical inspection of cluster distributions. PCA was not used for cluster determination or assignment.

### Step 2
**Cluster-specific correlation and network analysis.** Following cluster assignment, exploratory analyses were conducted to characterize cluster-specific patterns of association rather than to test predefined hypotheses or infer causality. TT levels were categorized into four data-driven quartiles (Q1–Q4), and analyses focused on relationships between TT strata and demographic, anthropometric, endocrine, inflammatory, metabolic, and renal variables within each cluster.

**Correlation analysis.** Pairwise correlations were calculated separately within each cluster using Pearson's correlation coefficient. Given the large number of correlations examined, analyses were treated as exploratory, with emphasis placed on consistency of overall patterns rather than individual $p$-values. Correlations with an absolute value

below 0.2 were excluded to reduce noise, and multiple testing correction was not applied.

**Network construction and summary.** Correlation matrices were reformatted into network representations to visualize multivariate association structures within clusters. Nodes represented individual variables (including TT quartile indicators), and edges represented correlations exceeding the predefined threshold. Network structure was summarized using eigenvector and betweenness centrality as descriptive measures of connectivity, without implying biological hierarchy or causal influence.

**Visualization and robustness.** All analyses were conducted using the same standardized and imputed dataset as in Step 1. Networks were visualized using force-directed layouts, with node size proportional to degree centrality and edge width reflecting correlation strength, to support qualitative comparison of network structure across clusters.

### Step 3

**External validation framework and analytical thresholds.** External validation was performed using data from the NHANES 2015–2016 cycle to examine whether biomarker-defined profiles identified in the primary cohort were associated with differences in clinically relevant outcomes in an independent population. This analysis was intended to assess consistency of multivariate physiological patterns rather than to establish causal relationships.

The NHANES dataset was restricted to male participants with available data on TT, CRP, serum creatinine, and self-reported cancer history. Cancer was selected as the validation outcome because it represents a clinically established, system-level endpoint reflecting long-term effects of endocrine status, inflammation, and organ function, and because it is consistently recorded in population-based datasets.

Biomarker-defined profiles were constructed using established clinical thresholds. TT < 300 ng/dL was interpreted as testosterone deficiency, serum creatinine >1.2 mg/dL as renal impairment, and CRP > 3.0 mg/L as elevated systemic inflammation. These thresholds were applied to classify individuals into profiles corresponding to those observed in the primary cohort.

Because of differences in sampling framework, population characteristics, and measurement protocols between the primary cohort and NHANES, analyses were limited to threshold-based classification and comparison of cancer prevalence. Clustering procedures and age-related trajectory analyses were not applied in NHANES.

### Statistical processing and software

All analyses were performed in Python using standard libraries (Pandas and NumPy). Continuous variables were summarized as median and interquartile range. Overall distributional differences across clusters were assessed using the Kruskal–Wallis test as a global screening procedure; post-hoc pairwise comparisons were not performed, consistent with the exploratory nature of the study. The singleton cluster was excluded from inferential analyses. Pairwise correlations were calculated using Pearson's correlation coefficient, and correlations with an absolute value below 0.2 were excluded to emphasize robust associations. Network graphs and scatter plots were generated using Matplotlib and NetworkX, with analyses focusing on overall structural patterns rather than hypothesis-driven testing. Code was developed and refined using ChatGPT (OpenAI, San Francisco, CA, USA) and executed in Google Colab (Google LLC, Mountain View, CA, USA).

## Results
### Step 1

**Unsupervised clustering of multivariate biomarker profiles.** The study encompassed a total of 5,854 patients, with ages ranging from 20 to 98 years and a mean age of $65.82 \pm 12.43$ years. Among them, 1,915 patients (32.71%) were aged 60–69 years, 1,959 (33.46%) were aged 70–79 years, 546 (9.33%) were aged 80–89 years, and 32 (0.55%) were aged 90 years or older. The BMI averaged $23.81 \pm 3.73$ kg/m². The mean serum total testosterone level was $4.43 \pm 2.75$ (available data; $N = 5854$). Hemoglobin (Hb) and hematocrit (Hct) averaged $13.95 \pm 1.69$ g/dL ($N = 5686$) and $41.17 \pm 4.68\%$ ($N = 5686$), respectively. Gonadotropins showed mean values of $7.78 \pm 7.95$ mIU/mL for luteinizing hormone ($N = 4611$) and $13.18 \pm 14.35$ mIU/mL for follicle-stimulating hormone ($N = 4491$). Metabolic markers included triglycerides of $145.87 \pm 93.42$ mg/dL ($N = 2435$) and high-density lipoprotein cholesterol of $51.56 \pm 14.89$ mg/dL ($N = 1938$). C-reactive protein averaged $0.44 \pm 1.61$ mg/dL ($N = 4833$).

### Hierarchical clustering reveals distinct patient groups

This section describes the use of hierarchical clustering as an exploratory approach to visualize overall similarity patterns among patients. Clinical indicators included age, height, body weight, BMI, Hb, Hct, albumin, aspartate aminotransferase (AST), alanine aminotransferase (ALT), alkaline phosphatase, creatinine (Cre), glucose, triglycerides, high-density lipoprotein cholesterol, low-density lipoprotein cholesterol, luteinizing hormone, follicle-stimulating hormone, CRP, and TT. After inclusion of subjects with complete age and BMI data, remaining missing values were handled by column-wise mean imputation, and all variables were standardized.

Pairwise similarity among subjects was assessed using Euclidean distance, and hierarchical clustering was performed using Ward linkage. The resulting dendrogram (Fig. 1) provides a qualitative overview of the global similarity structure across subjects.

This analysis was intended solely for exploratory visualization and did not aim to define cluster membership or determine the number of clusters. Formal cluster assignment and phenotypic characterization were conducted using K-means clustering, as described in subsequent sections.

### Clustering of male subjects using K-means

This section explains how the number of clusters was determined and how K-means clustering was applied. The elbow plot (Fig. 2) shows the within-cluster sum of squares (SSE) across different numbers of clusters, with a clear inflection around $K = 4$, beyond which further reductions in SSE become progressively smaller.

Although the average silhouette coefficient reached a marginal maximum at $K = 3$, the difference between $K = 3$ and $K = 4$ was modest. In contrast, examination of cluster centroids indicated that the $K = 4$ solution subdivided individuals with low TT and elevated C-reactive protein into two phenotypically distinct subgroups according to renal function. On this basis, $K = 4$ was selected to balance clustering metrics with clinical interpretability of the resulting phenotypes.

### Principal component analysis and visualization of clustering results

This section describes the preprocessing steps and clustering visualization using PCA. The dataset used in this analysis comprised male outpatients with multiple clinical variables, including age, BMI, and various blood test parameters. After applying the inclusion criterion requiring complete age and BMI data, the final analytical cohort consisted of 5854 subjects. Missing values in the remaining variables were subsequently imputed using column-wise mean imputation. After preprocessing, PCA and K-means clustering were performed for visualization and exploratory analysis (Fig. 3).

Following PCA, the data were projected onto the first two principal components (PC1 and PC2), which explained 18.31% and 11.47% of the total variance, respectively (cumulative 29.78%). Because these two components captured a limited proportion of the total variance, PCA was used exclusively for visualization and exploratory inspection. K-means clustering was then applied to the standardized data to classify individuals into four clusters ($K = 4$).

The results of the clustering were visualized in a two-dimensional scatter plot, where PC1 and PC2 were plotted as the axes (Fig. 3). Each

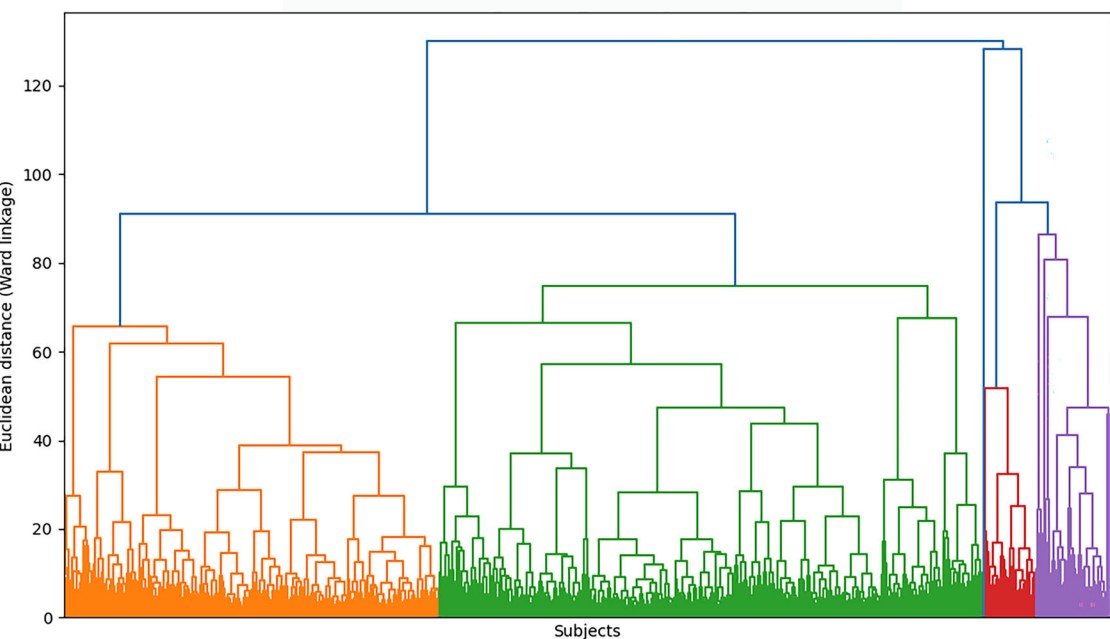

**Fig. 1 | Hierarchical clustering dendrogram of patient similarity (exploratory visualization).** Dendrogram based on hierarchical clustering with Ward linkage and Euclidean distance, using the standardized multivariate dataset after inclusion of subjects with complete age and body mass index data (N = 5854). Variables encompassing demographic, anthropometric, and biochemical parameters, with remaining missing values handled by column-wise mean imputation prior to standardization. The vertical axis corresponding to Euclidean distance under Ward linkage. Individual subject labels omitted for clarity. Visualization for qualitative inspection of global similarity structure rather than cluster determination.

cluster was represented by a different color to facilitate visual inspection, and the PC2 axis was scaled from −5 to 20 for visual clarity. As expected for a low-dimensional projection that captures only 29.78% of the total variance, the first two principal components did not yield clear visual separation among clusters. Accordingly, interpretation of cluster structure was based on analyses conducted in the full standardized feature space rather than on the PCA projection.

To assess the robustness of cluster assignment to alternative imputation strategies, KNN and iterative imputation were additionally evaluated. Cluster assignments remained broadly consistent across imputation methods (Adjusted Rand Index [ARI]: 0.72 for Mean vs. KNN, 0.71 for Mean vs. Iterative, and 0.69 for KNN vs. Iterative).

### Cluster characteristics

This section summarizes the clinical and biochemical characteristics of clusters derived from K-means clustering. Differences across demographic, anthropometric, inflammatory, renal, and hormonal variables were observed among clusters, as summarized in Table 1.

Cluster 0 comprised predominantly older individuals with lower body mass, elevated inflammatory markers, and impaired renal function. This cluster was characterized by lower Hb and albumin levels, markedly elevated gonadotropins, and reduced TT concentrations, reflecting a profile of combined endocrine, inflammatory, and renal alteration.

Cluster 1 included individuals of similar chronological age to Cluster 0 but exhibited lower levels of systemic inflammation and largely preserved renal function. TT concentrations were higher than those observed in Cluster 0, and gonadotropin levels were comparatively lower, indicating a more stable endocrine profile despite advanced age.

Cluster 2 represented a younger population with higher body mass and favorable metabolic characteristics. Inflammatory markers were low, renal function was preserved, and TT levels were relatively high compared with older clusters, suggesting a physiologically distinct profile associated with younger age.

Cluster 3 consisted of a single individual and was retained solely as an exploratory subgroup. Because meaningful summary statistics and between-group comparisons are not possible for a singleton cluster, it was excluded from the main descriptive table and inferential statistical analyses and is not interpreted further. Overall, these results indicate that K-means clustering identified subgroups with distinct distributions of clinical and biochemical characteristics across age, inflammatory status, renal function, and hormonal measures (Table 1). Statistical comparisons were interpreted as reflecting overall distributional differences among clusters rather than specific pairwise contrasts.

### Step 2

**Cluster-specific correlation and network analysis.** This section examines the associations between TT levels and physiological parameters within each of the four clusters identified by K-means clustering. TT levels were categorized into four quartiles using data-driven thresholds: Q1 (0.04–2.82 ng/mL), Q2 (2.83–4.42 ng/mL), Q3 (4.43–6.04 ng/mL), and Q4 (6.05–30.20 ng/mL). The analysis focused on relationships between TT and clinical variables including age, body weight, BMI, Hb, Hct, liver-related enzymes, and lipid-associated markers within each cluster.

In Cluster 0, TT levels tended to show inverse associations with age and adiposity-related measures, with relatively higher TT levels observed more frequently among younger individuals and lower BMI values. Associations between TT and liver-related enzymes were also observed in this cluster; specifically, TT showed weak but consistent negative correlations with AST and ALT in lower TT quartiles (e.g., AST $r = −0.14$ in Q1; ALT $r \approx −0.08$ in Q1–Q2), indicating modest inverse relationships between TT levels and hepatic enzyme activity.

In Cluster 1, age-related variation was accompanied by parallel changes in hematological and liver-associated parameters. TT levels in this cluster demonstrated coordinated patterns with Hb, Hct, and liver enzymes,

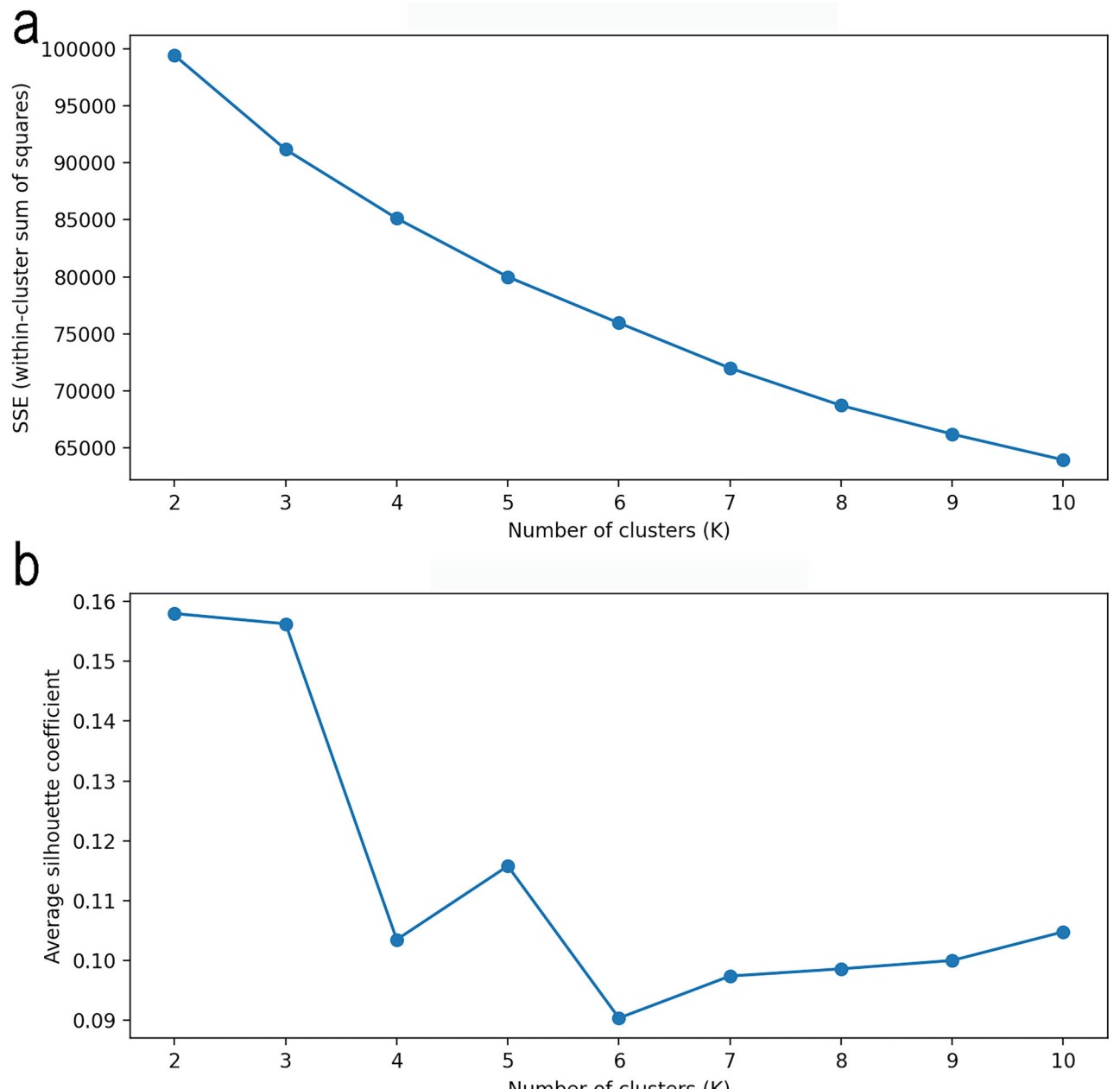

**Fig. 2 | Elbow and silhouette analyses for selection of the number of K-means clusters. a** Elbow plot of within-cluster sum of squares (SSE) across numbers of clusters ($K = 2–10$). A pronounced SSE reduction up to $K = 4$, followed by progressively smaller decreases at higher $K$ values, consistent with diminishing returns. **b** Average silhouette coefficients across $K$ values. Peak silhouette values at $K = 2–3$, lower values at $K = 4$, and a modest difference between $K = 3$ and $K = 4$. Overall assessment based on multiple criteria, including SSE behavior, silhouette distribution, and clinical separability of biomarker-defined phenotypes.

suggesting age-related physiological remodeling that may interact with testosterone regulation.

Cluster 2 represented a single-subject exploratory subgroup and was therefore not subjected to interpretive correlation analysis.

In Cluster 3, TT levels displayed heterogeneous associations across metabolic and inflammatory markers, without a single dominant correlation pattern. However, coordinated variation among liver-related enzymes was consistently observed across TT quartiles, suggesting preserved coupling of hepatic markers irrespective of testosterone level.

Detailed correlation coefficients, sample sizes, and statistical measures for each cluster and TT quartile are provided in Supplementary Table S3.

### Graph theory–based correlation structure analysis

This section describes cluster-specific correlation structures among clinical and biochemical variables using network representations. The analysis focuses on differences in correlation patterns and centrality patterns of variables within each cluster, without inferring causality or functional hierarchy.

Figure 4 presents correlation networks for selected clusters identified by K-means clustering. Network nodes correspond to individual variables and are labeled as "variable (cluster number)" (e.g., BMI(0)). Node size reflects relative connectivity (degree), and edges represent correlations exceeding a predefined threshold. These networks provide an exploratory overview of variable interrelationships within each cluster.

**Fig. 3 | Principal component analysis (PCA) and clustering visualization.** Distribution of subjects along the first two principal components, PC1 (18.31%) and PC2 (11.47%), from the standardized multivariate dataset ($N$ = 5854). Scatter plot with PC1 on the x-axis and PC2 on the y-axis. Points color-coded by cluster assignment: Cluster 0 (blue, $n$ = 2295), Cluster 1 (orange, $n$ = 528), Cluster 2 (singleton, $n$ = 1), and Cluster 3 (red, $n$ = 3030). PCA for visualization and exploratory inspection only; cluster assignment from K-means clustering in the full standardized feature space. PC2 axis range from −5 to 20.

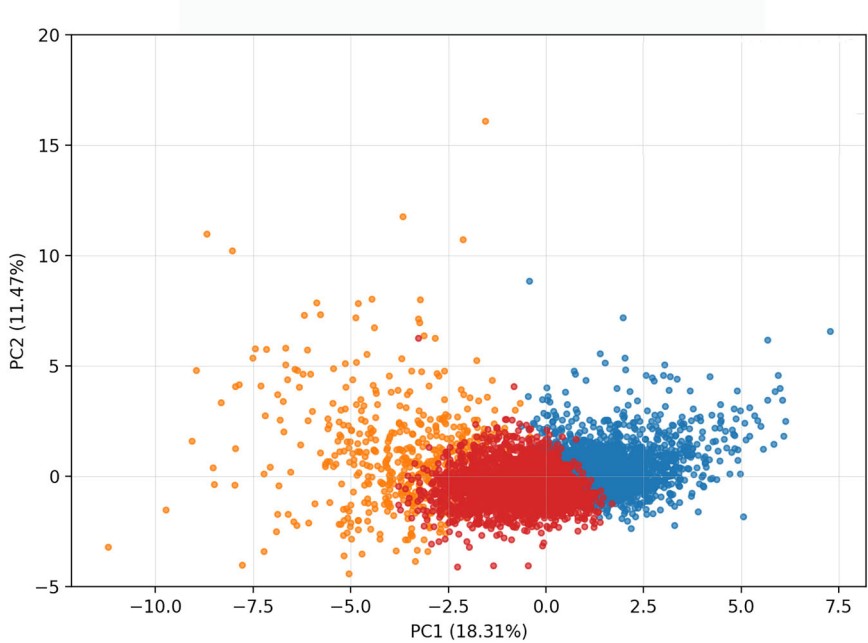

## Table 1 | Clinical and biochemical characteristics of the study population by K-means cluster (median [IQR])

| Variable | All ($n$ = 5854) | Cluster 0 ($n$ = 2295) | Cluster 1 ($n$ = 528) | Cluster 3 ($n$ = 3030) | Kruskal–Wallis $p^†$ |
|---|---|---|---|---|---|
| Age (years) | 68.00 (60.00–74.00) | 60.00 (52.00–67.00) | 75.00 (68.00–81.00) | 71.00 (66.00–76.00) | <0.001 |
| 60–69 years | 1915 (32.7%) | 869 (71.1%) | 98 (21.6%) | 948 (34.6%) | - |
| 70–79 years | 1959 (33.5%) | 327 (26.7%) | 203 (44.7%) | 1,429 (52.1%) | - |
| 80–89 years | 546 (9.3%) | 27 (2.2%) | 153 (33.7%) | 365 (13.3%) | - |
| Height (cm) | 166.70 (162.40–171.10) | 170.50 (166.80–174.60) | 163.70 (159.00–168.03) | 164.50 (160.80–168.00) | <0.001 |
| Body weight (kg) | 65.50 (58.52–73.00) | 74.40 (69.30–80.40) | 58.00 (51.25–65.62) | 60.90 (55.70–65.50) | <0.001 |
| BMI (kg/m²) | 23.56 (21.51–25.78) | 25.57 (23.77–27.85) | 21.69 (19.54–24.20) | 22.49 (20.63–24.15) | <0.001 |
| Hb (g/dL) | 14.20 (13.10–15.10) | 15.00 (14.30–15.80) | 10.90 (9.80–12.00) | 13.80 (12.90–14.50) | <0.001 |
| Hct (%) | 42.00 (39.00–44.90) | 44.00 (41.60–46.20) | 33.00 (30.00–36.20) | 40.60 (38.20–43.00) | <0.001 |
| Albumin (g/dL) | 4.10 (3.80–4.30) | 4.30 (4.10–4.50) | 3.40 (3.10–3.80) | 4.00 (3.80–4.30) | <0.001 |
| Cre (mg/dL) | 0.81 (0.69–0.98) | 0.80 (0.70–0.90) | 0.99 (0.86–1.20) | 0.81 (0.70–0.94) | <0.001 |
| CRP (mg/dL) | 0.10 (0.10–0.30) | 0.10 (0.10–0.20) | 0.70 (0.30–2.20) | 0.10 (0.10–0.20) | <0.001 |
| TT (ng/mL) | 4.43 (2.90–5.90) | 4.34 (3.00–5.60) | 2.90 (1.70–4.30) | 4.69 (3.30–6.10) | <0.001 |
| LH (mIU/mL) | 6.30 (4.20–9.30) | 5.20 (3.80–7.10) | 12.40 (7.60–19.60) | 6.70 (4.80–10.20) | <0.001 |
| FSH (mIU/mL) | 9.10 (5.90–14.20) | 7.10 (5.10–10.10) | 19.60 (12.40–30.30) | 10.20 (7.10–15.40) | <0.001 |

Clinical, anthropometric, inflammatory, renal, and hormonal characteristics across clusters derived from K-means clustering. Values are presented as median and interquartile range (IQR). Comparisons across clusters were assessed using the Kruskal-Wallis test as a global screening for overall distributional differences among Clusters 0, 1, and 3; post-hoc pairwise comparisons were not performed. The singleton cluster (Cluster 2) was excluded from inferential analyses and is reported separately as an exploratory subgroup.
*BMI* body mass index, *Hb* hemoglobin, *Hct* hematocrit, *Alb* albumin, *Cre* creatinine, *CRP* C-reactive protein, *LH* luteinizing hormone, *FSH* follicle-stimulating hormone, *TT* serum total testosterone.
†$p$ values derived from the Kruskal -Wallis test.

For reference, representative Spearman correlation coefficients in Cluster 0 included inverse associations between TT and liver enzymes (e.g., AST $r ≈ −0.14$; ALT $r ≈ −0.08$; see Supplementary Table S3 for full coefficients), whereas correlations in other clusters were generally weaker or more heterogeneous.

In Cluster 0, higher relative centrality was observed among variables related to TT status and body composition, including Q1_lowTT, TT(0), and BMI(0). This cluster exhibited a compact correlation structure centered on hormonal and metabolic measures.

Cluster 1 showed a more distributed correlation pattern, with hematological and renal markers such as Hct(1) and Cre(1) displaying higher centrality relative to other variables. Overall network density appeared lower than in Cluster 0.

Cluster 2 demonstrated prominent connectivity among metabolic and lipid-related variables, including Q1_lowTT(2), BMI(2), and HDL-C(2), forming a localized subnetwork within the broader correlation structure.

In Cluster 3, variables related to body composition and testosterone, including BMI(3) and TT(3), showed higher relative connectivity,

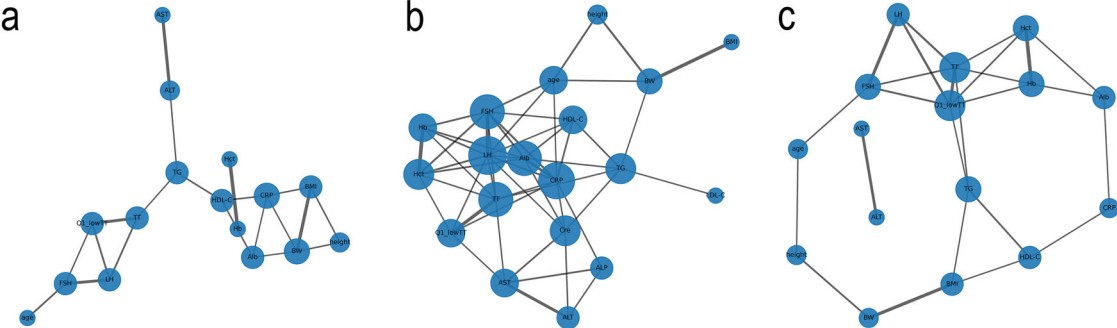

**Fig. 4 | Cluster-specific correlation network structures of physiological variables.** Cluster-specific correlation network representations of physiological variables across selected clusters identified by K-means clustering. **a**, **b**, **c** corresponding to Cluster 0, Cluster 1, and Cluster 3, respectively. Network nodes representing clinical and biochemical variables, including age, height, body weight (BW), body mass index (BMI), hemoglobin (Hb), hematocrit (Hct), albumin (Alb), liver-related enzymes (AST, ALT, ALP), creatinine (Cre), triglycerides (TG), high-density lipoprotein cholesterol (HDL-C), luteinizing hormone (LH), follicle-stimulating hormone (FSH), C-reactive protein (CRP), total testosterone (TT), and the low-TT indicator (Q1_lowTT). Edges indicating Spearman correlations above a predefined threshold, with edge width proportional to correlation strength. Node size representing relative connectivity (degree). Exploratory visualization of cluster-level correlation structures.

accompanied by distinct subnetwork formations involving liver-related markers. Across clusters, the relative connectivity of Q1_lowTT differed, appearing more central in Clusters 0 and 2 than in Clusters 1 and 3. These observations highlight heterogeneity in correlation structures across clusters and support the exploratory characterization of cluster-specific physiological profiles.

### Analysis of age-related changes and breakpoints in testosterone, CRP, and creatinine

This section examines age-related patterns in TT, CRP, and Cre across the entire analytical cohort and within Clusters 0 and 3, which predominantly consist of older individuals. Because physiological aging may involve non-linear changes, both correlation-based analyses and piecewise regression were applied to characterize age-associated trends without inferring causality.

In the overall cohort (Fig. 5a–c), TT demonstrated a clear age-associated decline. TT levels remained relatively stable through early adulthood and midlife, followed by a marked decrease in older age, reflected by a significant negative correlation with age (Spearman $r < 0$, $p < 0.001$). In contrast, CRP showed no consistent monotonic trend with age, although a modest inverse association between TT and CRP was observed across the population (Spearman $r < 0$, $p < 0.001$). Creatinine levels exhibited minimal correlation with age in the overall cohort, indicating relative stability across the adult lifespan.

Analyses restricted to Clusters 0 and 3 (Fig. 5d–f) revealed distinct age-related patterns compared with the overall population. Piecewise regression identified an earlier inflection point for TT, with a breakpoint in the early 50 s, followed by a sustained decline at older ages. Breakpoints were estimated independently for each biomarker, and dashed vertical lines in Fig. 5d–f indicate biomarker-specific inflection points derived from piecewise linear regression. Within these clusters, the inverse association between TT and CRP was stronger than in the overall cohort (Spearman $r < 0$, $p < 0.001$), indicating closer coupling between hormonal status and inflammatory markers in older individuals.

For creatinine, piecewise analysis in Clusters 0 and 3 indicated a change in slope at later ages, with decreasing values before the breakpoint and relative stabilization thereafter. A modest positive correlation between TT and creatinine was observed within these clusters (Spearman $r > 0$, $p < 0.01$), suggesting an association between testosterone levels and renal function markers in this older subgroup, although the magnitude of this relationship remained limited.

Together, these findings indicate that age-related changes in TT, CRP, and creatinine differ between the overall population and older-dominant clusters. The presence of non-linear patterns and earlier inflection points in Clusters 0 and 3 supports heterogeneity in aging-associated biomarker trajectories and underscores the value of cluster-specific analyses for exploratory characterization of aging-related physiological profiles.

### Step 3

**External validation using cancer prevalence.** External validation was performed using data from the NHANES 2015–2016 cohort to examine whether biomarker-defined profiles corresponding to Clusters 0 and 3 were associated with distinct cancer prevalence patterns in an independent population. The analysis included male participants ($n = 2,463$) with complete data on TT, C-reactive protein, serum creatinine, and self-reported cancer history.

Cluster definitions were aligned with those derived from the Japanese cohort. Cluster 0 was defined by impaired renal function (serum creatinine > 1.2 mg/dL), elevated CRP (>3 mg/L), and low TT (<300 ng/dL). Cluster 3 was defined by preserved renal function with elevated CRP and low TT. Participants not meeting either definition were classified as a Healthy reference group.

Site-specific lifetime and recent cancer prevalence across these biomarker-defined groups is summarized in Table 2. Overall differences in cancer prevalence across groups were observed, with the highest burden consistently identified in Cluster 0. In contrast, Cluster 3 showed a cancer distribution that differed in composition from Cluster 0 but did not consistently exceed that of the Healthy group across cancer categories.

Analyses of cancers diagnosed within the preceding 5 years demonstrated a similar pattern, with higher recent cancer prevalence in Cluster 0 compared with the Healthy group. Differences between Cluster 3 and the Healthy group were less consistent across cancer categories.

Taken together, these findings support external consistency of the biomarker-defined high-risk profile corresponding to Cluster 0, while Cluster 3 exhibited a distinct but more heterogeneous cancer pattern relative to the Healthy reference group. These patterns may relate to chronic inflammation, endocrine dysregulation, and renal function, without implying causal relationships.

### Discussion

This study explored heterogeneity in aging-related physiological states by examining multivariate biomarker patterns centered on TT in a large clinical cohort. Rather than treating aging as a uniform or linear process, the analyses revealed that endocrine, inflammatory, metabolic, and renal markers co-occur in distinct configurations across individuals[54,55]. These findings support the view that biological aging is characterized by substantial inter-individual variability[56] and that such variability may not be adequately captured by chronological age or single biomarkers alone[57,58].

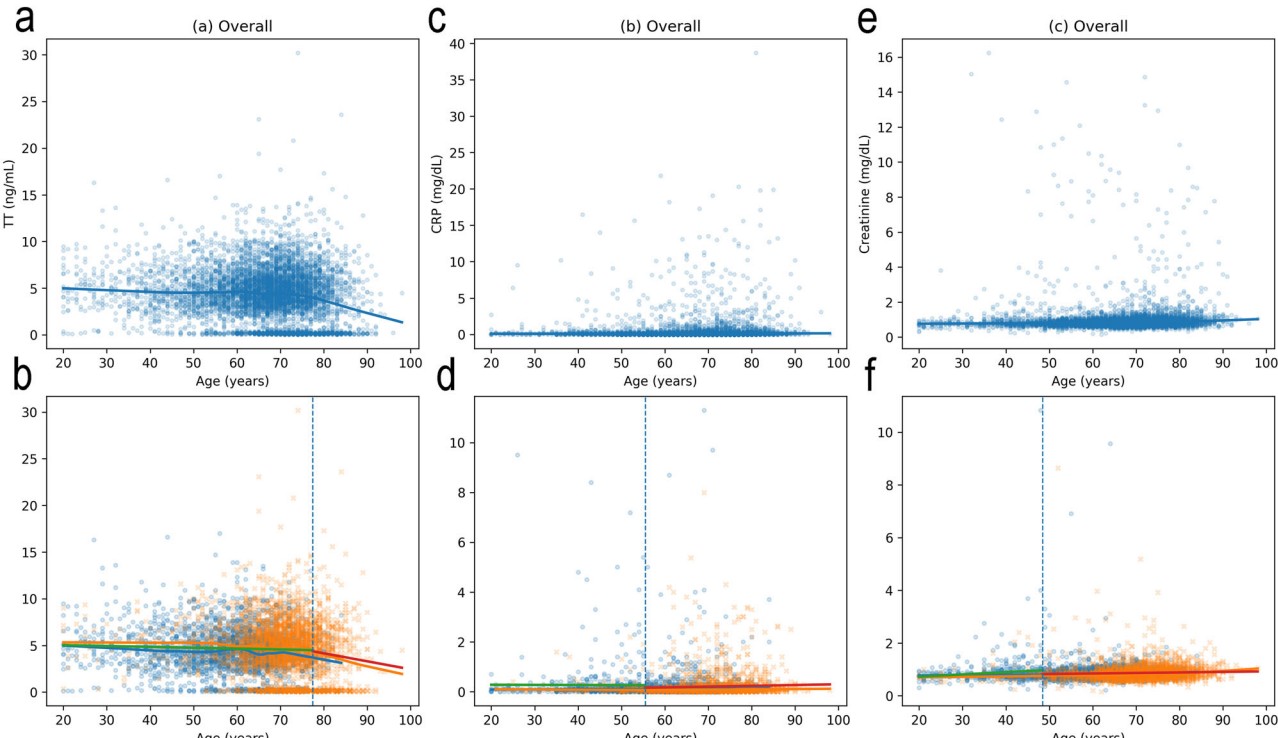

**Fig. 5 | Age-related patterns of total testosterone, C-reactive protein, and creatinine in the overall cohort and in Clusters 0 and 3.** X-axis (all panels): Age (years). Y-axis: **a**, **d** TT (ng/mL); **b**, **e** CRP (mg/dL); **c**, **f** Cre (mg/dL). **a–c** overall cohort; **d–f**, Clusters 0 and 3. Symbols in **d–f**: Cluster 0 (circles), Cluster 3 (crosses). Solid lines: LOESS smoothing curves. Dashed vertical lines in **d–f**: breakpoints from piecewise linear regression in Clusters 0 and 3. Abbreviations: TT total testosterone, CRP C-reactive protein, Cre creatinine.

First, the results highlight heterogeneity in aging-related physiological profiles. Clustering analysis identified subgroups with distinct combinations of TT levels, inflammatory status, and renal function, demonstrating that similar chronological ages may correspond to markedly different physiological states[57]. In particular, one cluster was characterized by the co-occurrence of low TT, elevated C-reactive protein, and impaired renal function[59]. This pattern is best interpreted as a descriptive configuration of biomarker distributions rather than as a discrete disease entity or a definitive aging phenotype. The cross-sectional nature of the data precludes conclusions regarding temporal sequence or causality[60]. Nonetheless, the observed clustering underscores that aging-related physiological changes do not progress uniformly and may instead manifest as different combinations of system-level alterations across individuals[57,61,62].

Second, the use of unsupervised clustering and network-based analysis provides insight into structural differences among physiological variables without implying mechanistic directionality[63,64]. Network representations demonstrated that the relative connectivity of TT, body composition, inflammatory markers, and renal indicators differed across clusters, suggesting that the role of TT varies depending on the surrounding physiological context. Importantly, network centrality reflects statistical relationships rather than biological hierarchy or causal influence[65]. From this perspective, clustering and network analysis should be viewed as descriptive tools that facilitate characterization of multivariate physiological structure[66,67]. These approaches complement, rather than replace, traditional hypothesis-driven analyses and are particularly suited for exploratory studies aimed at mapping heterogeneity rather than testing specific causal models[68,69].

Third, the external validation using cancer prevalence supports the potential clinical relevance of the biomarker-defined profiles. Cancer was selected as an outcome because it represents a clinically established, system-level endpoint that integrates long-term influences of inflammation, endocrine dysregulation, and organ function, and because cancer history is consistently documented in population-based datasets[70,71]. Individuals meeting the biomarker criteria corresponding to one cluster exhibited higher cancer prevalence in an independent dataset, whereas other profiles showed different or less pronounced patterns. This analysis was not intended to establish causal links between physiological patterns and cancer development[60]. Instead, it provides evidence of consistency across datasets, suggesting that multivariate biomarker configurations identified in clinical data may correspond to meaningful differences in health outcomes beyond chronological age alone[72–75].

Finally, these findings have implications for future research on aging and risk stratification. The results suggest that evaluating combinations of endocrine, inflammatory, and renal markers may offer a more nuanced description of aging-related physiological states than age-based frameworks alone[54–59,61,62]. Network-informed, multivariate approaches may help identify subgroups for closer monitoring or for hypothesis generation in longitudinal studies[63–67,69,70]. Future work incorporating repeated measurements over time will be essential to determine whether the observed biomarker configurations correspond to distinct aging trajectories[76,77], differential disease risk, or functional outcomes. Integration with additional biological domains, such as genomic or lifestyle factors, may further refine understanding of heterogeneity in aging processes[61,62].

There are some limitations that should be acknowledged. The cross-sectional design limits inference regarding temporal relationships and causality[60]. The study population consisted exclusively of Japanese men undergoing clinical testing, which may restrict generalizability to other populations or to women. Missing data were addressed through imputation to preserve cohort size, and although clustering stability was evaluated across imputation methods, some influence of data handling choices cannot be excluded[60]. Network analyses were exploratory and dependent on variable selection and correlation thresholds[48,49,66,67]. Finally, cancer prevalence was used as an external validation outcome based on availability and robustness, but other clinically relevant outcomes were not examined[53,72].

**Table 2 | External validation of cancer risk across clusters defined by TT, inflammation, and kidney function in the NHANES 2015–2016 cohort**

|  | Category | Cluster 0 | Cluster 3 | Healthy |
|---|---|---|---|---|
| Lifetime cancer prevalence (%) | Prostate cancer | 8.96 | 4.56 | 3.33 |
|  | Lung cancer | 2.99 | 0.76 | 0.09 |
|  | Bladder cancer | 4.48 | 0.38 | 0.38 |
|  | Colorectal cancer (colon + rectal) | 0.00 | 0.76 | 0.70 |
|  | Skin cancer (non-melanoma + unspecified) | 2.99 | 3.80 | 2.67 |
|  | Melanoma (malignant melanoma) | 1.49 | 0.38 | 0.98 |
|  | Other cancers (remaining) | 7.46 | 2.28 | 1.55 |
| Recent cancer prevalence (diagnosed within 5 years, %) | Prostate cancer | 8.70 | 10.00 | 1.48 |
|  | Lung cancer | 4.35 | 1.67 | 0.08 |
|  | Bladder cancer | 4.35 | 1.67 | 0.25 |
|  | Colorectal cancer (colon + rectal) | 2.17 | 0.00 | 0.25 |
|  | Skin cancer (non-melanoma + unspecified) | 4.35 | 6.67 | 0.89 |
|  | Melanoma | 4.35 | 0.00 | 0.42 |
|  | Other cancers (remaining) | 10.85 | 5.00 | 0.52 |

Percentages: proportion of individuals within each biomarker-defined group for each cancer category.
Upper section: lifetime cancer prevalence (self-reported history).
Lower section: cancers diagnosed within the preceding 5 years.
Group definitions: Cluster 0 (kidney impairment, low TT, elevated CRP); Cluster 3 (preserved kidney function, low TT, elevated CRP); Healthy reference group (criteria not met).
Cancer categories: consolidation for major site-specific patterns relevant to biomarker-defined clusters.
Colorectal cancer: colon and rectal cancer.
Skin cancer: non-melanoma and unspecified skin cancers.
Other cancers (remaining): cancer types not listed individually.

Despite these limitations, the study demonstrates that multivariate, exploratory analyses can reveal structured heterogeneity in aging-related physiological profiles[54–59,61–63,67]. Such approaches may complement conventional age-based perspectives by emphasizing system-level configurations and inter-individual variation, providing a foundation for future longitudinal and translational investigations[54–59,61,62,75].

## Conclusion
In this study, unsupervised clustering of testosterone and related endocrine, metabolic, inflammatory, and renal biomarkers identified heterogeneous aging-related physiological profiles in men. The results indicate that variations in these biomarkers are organized into cluster-specific network patterns, suggesting that testosterone functions not as an isolated hormonal marker but as part of an integrated physiological system interacting with multiple biological domains.

Although the present analysis is cross-sectional and exploratory and does not allow causal or temporal inference, it demonstrates that data-driven phenotyping can provide complementary insights beyond conventional clinical classifications. Future studies incorporating longitudinal data or interventional designs will be required to clarify the clinical relevance and potential predictive value of these aging-related

## Data availability
All input datasets used in this study are available from the cited references. The clinical and metabolic datasets generated during this work are available at the link below. The dataset includes detailed clinical and metabolic information on 5877 male patients, of whom 5854 individuals with complete age and BMI data were included in the present analyses. The data file is stored in .csv format (total file size: 643 KB). The source data underlying the figures and tables in this study are publicly available in the Zenodo repository at https://doi.org/10.5281/zenodo.18845986.

## Code availability
All computational analyses were performed in Python. The complete codebase and documentation to reproduce the analytical workflow are available on GitHub: https://github.com/Curiosity-Mars/testosterone. The repository includes scripts for data processing, statistical analysis, and visualization used in this study.

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

## Acknowledgements

The authors dedicate this work to the memory of the late Professor Yoshiaki Kumamoto, Honorary President of the Japanese Society for Men's Health, whose pioneering contributions elevated testosterone measurement in Japan to a scientific discipline and whose mentorship profoundly influenced our work.

## Author contributions

N.O. and S.H. contributed equally to all aspects of the study, from the initial draft to the final manuscript. N.O. and S.H. jointly conceived the study, conducted the analyses, interpreted the results, and prepared all figures and the manuscript.

## Competing interests

The authors declare no competing interests. This study was approved by the Juntendo University Hospital Research Ethics Board (Approval number: H19-0128). The study was conducted in accordance with the principles outlined in the Declaration of Helsinki.
