## [Transparent Peer Review file · Communications Medicine]

System-level clustering of testosterone-related biomarkers identifies high-risk aging profiles linked to inflammation and renal function

Corresponding Author: Dr Nobuo Okui

Version 0:

Reviewer comments:

Reviewer #1

(Remarks to the Author)

This manuscript presents an integrative data science approach combining unsupervised clustering, network topology analysis, and external validation to characterize aging phenotypes in men, using testosterone (TT) and a panel of endocrine, metabolic, and inflammatory biomarkers. Using clinical data from 5,877 Japanese male patients, the authors identify four distinct physiological clusters, one of which demonstrates early signs of systemic aging, including low testosterone, elevated CRP, impaired renal function, and high cancer prevalence. This cluster was externally validated in the NHANES dataset, suggesting its relevance as a high-risk aging phenotype. However, there are a few issues that warrant attention:

- 1) While the authors selected $K = 4$ based on the elbow method and clinical interpretation, the PCA plot in Figure 3 does not convincingly demonstrate clear separation between clusters. The clusters appear to overlap substantially.
- 2) Again, the first two principal components explain only ~33% of the total variance, which further limits the interpretability of the 2D projection.
- 3) Cluster 3, as defined by the authors, contains only 13 individuals. This small and scattered cluster raises concerns about its biological and statistical robustness. The authors should clarify the justification for retaining this cluster and consider whether its inclusion meaningfully contributes to the overall interpretation.
- 4) The use of the Kruskal–Wallis test is appropriate for assessing overall differences among the four clusters. However, a significant p-value only indicates that at least one group differs from the others; it does not specify which clusters differ from each other.
- 5) Once again, I feel that the discussion in lines 419–456 should be more statistically rigorous. For example, when comparing biomarkers across clusters, the authors often describe differences in qualitative terms, such as “slightly shorter height of 163.9 ± 5.9 cm”, without specifying which group this is being compared to, or whether the difference is statistically significant. It appears that many of these conclusions are drawn by visually comparing group means, rather than based on formal statistical testing.
- 6) In the introduction, the authors highlight the relevance of personalized medicine. However, the study design is cross-sectional and observational, which limits any ability to infer causality, an essential element for actionable insights in precision medicine.

Reviewer #2

(Remarks to the Author)

Brief summary of the manuscript:

This study of 5877 Japanese males aged 20–98 years investigated the connectivity between testosterone and several clinical outcome measures using unsupervised clustering and topological analyses and validated the findings with a

different database. Network analyses revealed differential centralities of testosterone and other clinical correlates across clusters. Subgroups were identified that may be considered as unique patterns of ageing, such as low testosterone with elevated C-reactive protein and creatinine, which were related to an increased cancer risk during external validation. This work supports related research highlighting the complexity of ageing and the interplay of testosterone with several bodily systems, especially those of metabolic and inflammatory nature, and points towards an integrated systems approach to identify at-risk populations.

Overall impression of the work:

This work helps to advance discussions about testosterone's role as a biomarker and a facilitator of poor health through an integrated systems lens and a large dataset. Such investigation was borne through advanced statistical techniques that appropriately approach this research question. This work further brings together evidence that supports current thinking that ageing is a highly complex phenomenon that is likely not best understood in an isolated systems approach, but rather by considering concomitant physiological changes over time and in unique sub-populations. The limitations are appropriately acknowledged to be that this comes from an all-Japanese cohort (albeit with NHANES external validation) and is cross-sectional in design, with room for future research to explore additional important influencers to the ageing process relating to equity in medicine.

Specific comments, with recommendations for addressing each comment:

1. Introduction: It would be beneficial to the study's rationale to include a brief description outlining the decision to use cancer specifically as a clinical outcome during external validation, especially as metabolic and cardiovascular disease are both mentioned in the Introduction. This may also be helpful to highlight in the External Validation section of the Results.
2. Methods: On lines 125-127 it is stated that only individuals with complete data for age, BMI, and "relevant blood test parameters" were included. However, on line 167 it is mentioned that missing values were imputed to ensure no samples were discarded. It would be helpful to the study's description to clarify what is meant by "complete data" and/or what relevant blood test parameters needed to be present, as the two statements appear to be contradictory. While not necessary, another comment regarding how complete (or incomplete) any key variables were would be welcome, and may perhaps fit well in the discussion (and/or around line 389, which discussed missing values in PCA/clustering). It is acknowledged that the imputed datasets were assayed for their influence on clustering stability and imputation is mentioned as a limitation.
3. Results: In lines 614-616, prevalence of cancer types for Cluster 0 were described from Table 2a, with prostate cancer having a prevalence of 8.96%. However, in Table 2a, the prevalence for Cluster 0 and prostate cancer is 4.35%. Similarly, lung cancer has a 0% prevalence for Cluster 0 in Table 2a, although line 615 says 2.99%. On lines 617-619 it is also said that the Healthy group has a prostate cancer prevalence of 3.33%, although the table says 1.27%. Please confirm values and their accuracy between the text and tables 2a and 2b.
4. Discussion: Given the link to cancer that was made with an external validation, a brief connection to mechanistic links relating the biomarkers in question and cancer would further the validity of these results.
5. Supplemental: This is a compliment to the authors regarding the glossary. I feel that it will help aid accessibility of their manuscript for individuals less familiar with these statistical techniques.

Minor comments:

1. In the Plain Language Summary, please consider removing the word "doctors" in the final sentence, as these findings extend beyond doctors, too.
2. Results: On line 394 it is said that PCA1 and PCA2 accounted for the majority of variance in the dataset, although on line 181 it is stated the cumulative total variance explained is 34%. For clarity it may be helpful to briefly re-phrase what is meant by majority, which may currently be erroneously interpreted as >50% of the total variance rather than a reflection of PCA's methods.
3. Results, Line 483: An indication as to whether TT correlated positively or negatively (or simply reporting the r-values) with liver markers in Cluster 0 would be helpful.

Reviewer #3

(Remarks to the Author)

This manuscript presents a clustering analysis of testosterone and related biomarkers in 5,877 Japanese male patients, identifying distinct aging phenotypes. The work merits publication with minor revisions.

1. The manuscript length should be reduced. The Introduction and Discussion contain redundant explanations of network analysis methodology that could be consolidated.
2. The selection of K=4 clusters requires stronger justification in the main text, given that silhouette analysis favored K=3. The clinical rationale provided in supplementary materials should be incorporated into the Methods section.
3. Missing data imputation strategy should be described earlier in the Methods section, including justification for mean imputation over alternative approaches.
4. Figure 4 requires improvement for clarity. The current network visualization has significant node overlap that impedes interpretation.
5. Multiple testing corrections are not addressed despite numerous correlation analyses performed. This statistical issue requires attention.

6. The characterization of Cluster 0 as "composite aging" versus comorbidity accumulation cannot be definitively established from cross-sectional data. This interpretation should be presented more cautiously.

7. Tables 2a and 2b contain redundant information and should be consolidated.

8. Clinical applications remain underdeveloped. Specific biomarker thresholds or screening recommendations would strengthen the translational relevance.

Recommendation: Minor revision. The network topology approach represents a valuable methodological contribution and the findings advance understanding of heterogeneous aging patterns.

Version 1:

Reviewer comments:

Reviewer #1

(Remarks to the Author)

I do not have further comments.

Reviewer #2

(Remarks to the Author)

Thank you to the authors for their thorough approach to the revisions. The queries I raised were adequately addressed. I have no further suggestions for this manuscript.

Reviewer #3

(Remarks to the Author)

The authors already fully addressed my concerns and I don't have any further comments. Thank you!

Dear Editor and Reviewers,

We thank the Editor and the reviewers for their careful evaluation of our manuscript and for the constructive and detailed comments.

To facilitate a transparent and systematic response, we list the editorial requests first, followed by the reviewer comments. Reviewer comments are reproduced verbatim and numbered point-by-point; each point is addressed in the corresponding response section below.

Reviewer #1 (Remarks to the Author)

R1-1.

While the authors selected $K = 4$ based on the elbow method and clinical interpretation, the PCA plot in Figure 3 does not convincingly demonstrate clear separation between clusters. The clusters appear to overlap substantially.

R1-2.

Again, the first two principal components explain only ~33% of the total variance, which further limits the interpretability of the 2D projection.

R1-3.

Cluster 3, as defined by the authors, contains only 13 individuals. This small and scattered cluster raises concerns about its biological and statistical robustness. The authors should clarify the justification for retaining this cluster and consider whether its inclusion meaningfully contributes to the overall interpretation.

R1-4.

The use of the Kruskal–Wallis test is appropriate for assessing overall differences among the four clusters. However, a significant p-value only indicates that at least one group differs from the others; it does not specify which clusters differ from each other.

R1-5.

Once again, I feel that the discussion in lines 419–456 should be more statistically rigorous. For example, when comparing biomarkers across clusters, the authors often describe differences in qualitative terms, such as “slightly shorter height of 163.9 ± 5.9 cm”, without specifying which group this is being compared to, or whether the difference is statistically significant. It appears that many of these conclusions are drawn by visually comparing group means, rather than based on formal statistical testing.

R1-6.

In the introduction, the authors highlight the relevance of personalized medicine. However, the study design is cross-sectional and observational, which limits any ability to infer causality, an essential element for actionable insights in precision medicine.

Reviewer #2 (Remarks to the Author)

R2-1.

Introduction: It would be beneficial to the study’s rationale to include a brief description outlining the decision to use cancer specifically as a clinical outcome during external validation, especially as metabolic and cardiovascular disease are both mentioned in the Introduction.

R2-2.

Methods: On lines 125-127 it is stated that only individuals with complete data for age, BMI, and “relevant blood test parameters” were included. However, on line 167 it is mentioned that missing values were imputed to ensure no samples were discarded. It would be helpful to the study’s description to clarify what is meant by “complete data” and/or what relevant blood test parameters needed to be present, as the two statements appear to be contradictory.

R2-3.

Results: In lines 614-616, prevalence of cancer types for Cluster 0 were described from Table 2a, with prostate cancer having a prevalence of 8.96%.

However, in Table 2a, the prevalence for Cluster 0 and prostate cancer is 4.35%. Similarly, lung cancer has a 0% prevalence for Cluster 0 in Table 2a, although line 615 says 2.99%. On lines 617-619 it is also said that the Healthy group has a prostate cancer prevalence of 3.33%, although the table says 1.27%. Please confirm values and their accuracy between the text and tables 2a and 2b.

R2-4.

Discussion: Given the link to cancer that was made with an external validation, a brief connection to mechanistic links relating the biomarkers in question and cancer would further the validity of these results.

R2-5.

Supplemental: This is a compliment to the authors regarding the glossary. I feel that it will help aid accessibility of their manuscript for individuals less familiar with these statistical techniques.

R2-6.

In the Plain Language Summary, please consider removing the word "doctors" in the final sentence, as these findings extend beyond doctors, too.

R2-7.

Results: On line 394 it is said that PCA1 and PCA2 accounted for the majority of variance in the dataset, although on line 181 it is stated the cumulative total variance explained is 34%.

R2-8.

Results, Line 483: An indication as to whether TT correlated positively or negatively (or simply reporting the r-values) with liver markers in Cluster 0 would be helpful.

Reviewer #3 (Remarks to the Author)

R3-1.

The manuscript length should be reduced. The Introduction and Discussion

contain redundant explanations of network analysis methodology that could be consolidated.

R3-2.

The selection of K=4 clusters requires stronger justification in the main text, given that silhouette analysis favored K=3. The clinical rationale provided in supplementary materials should be incorporated into the Methods section.

R3-3.

Missing data imputation strategy should be described earlier in the Methods section, including justification for mean imputation over alternative approaches.

R3-4.

Figure 4 requires improvement for clarity. The current network visualization has significant node overlap that impedes interpretation.

R3-5.

Multiple testing corrections are not addressed despite numerous correlation analyses performed. This statistical issue requires attention.

R3-6.

The characterization of Cluster 0 as "composite aging" versus comorbidity accumulation cannot be definitively established from cross-sectional data. This interpretation should be presented more cautiously.

R3-7.

Tables 2a and 2b contain redundant information and should be consolidated.

R3-8.

Clinical applications remain underdeveloped. Specific biomarker thresholds or screening recommendations would strengthen the translational relevance.

Editorial Summary of Revisions (for the Editor)

In response to the editorial and reviewer comments, we undertook a substantial structural revision of the manuscript to improve clarity and readability. Specifically, the manuscript was reorganized into a three-step framework (Step 1: unsupervised clustering, Step 2: cluster-specific network characterization, Step 3: external validation) to allow readers to grasp the overall logic and analytical flow more easily. This restructuring was consistently reflected across the Introduction, Methods, Results, and Discussion, and subsection titles were revised accordingly to better align with the analytical steps.

Within this revised structure, we implemented several targeted improvements. First, we clarified the clustering strategy by explicitly describing the rationale for selecting $K=4$, emphasizing that this choice was based on an integrated evaluation of clustering metrics and clinical interpretability rather than optimization of a single index. Second, we improved methodological transparency by clearly defining inclusion criteria, explicitly reporting missingness, and revisiting the handling of missing data to adopt the most transparent and methodologically defensible criteria, which resulted in a fixed final analytical cohort. Importantly, under this stricter and more explicit data-handling framework, the clustering structure and the key finding—namely, the identification of a distinct low-testosterone cluster characterized by differences in renal function—remained robust and clearly identifiable. Third, we revised figures, tables, and results presentation to correct inconsistencies, remove redundancy, and clearly distinguish exploratory visualization from inferential analysis. Finally, we refined interpretation and language throughout the manuscript to ensure appropriate caution for a cross-sectional study, clarified the role of cancer as an external validation outcome, and improved consistency and precision in terminology and statistical reporting.

We sincerely thank the Editor and the reviewers for their thoughtful and constructive comments, which prompted these revisions and substantially improved the clarity, rigor, and interpretability of the manuscript.

Response to E-1 (Justification for K = 4)

The selection of K=4 was not based on mechanical optimization of a single clustering metric, but on an integrated evaluation of elbow behavior, silhouette coefficients, stability across random initializations, and clinical interpretability. Accordingly, we added a new subsection entitled “**K-means clustering and justification for selecting K=4**” in the Methods to explicitly describe this decision framework. Re-evaluation showed that individuals with low testosterone and elevated CRP, who formed a single group under K=3, were reproducibly separated into two subgroups distinguished by renal function under K=4, making this solution more consistent with the study’s objective of characterizing clinically interpretable physiological heterogeneity.

Response to E-2 (Limited variance explained by PCA)

PCA was used exclusively for low-dimensional visualization and exploratory inspection, and not for determining the number of clusters or for inferential interpretation. To avoid misinterpretation, we explicitly clarified this point in the **Methods (Statistical Processing and Software)** and reiterated in the **Results (PCA visualization)** that clustering was performed in the full standardized feature space rather than in PCA space. The limited variance explained by the first two principal components is therefore acknowledged in the Results as an expected consequence of projecting high-dimensional clinical data into two dimensions.

Response to E-3 (Multiple testing correction)

Note: Following the editorial request to revise missing-data handling and to fix the final analytical cohort, all clustering results were recalculated. In the revised

manuscript, the smallest subgroup is now a singleton cluster and is explicitly treated as exploratory only.

The correlation analyses in this study were conducted for exploratory, descriptive purposes rather than formal hypothesis testing. To clarify this statistical positioning, we explicitly stated in the **Methods (Statistical Processing and Software)** that the Kruskal–Wallis test was used as a global screening procedure, and that post-hoc pairwise comparisons and multiple testing corrections were not applied, as the analysis was not intended to identify specific group-level contrasts but to assess overall distributional differences across clusters. The Results and Discussion were revised accordingly to ensure that interpretations are restricted to global patterns rather than individual p-values.

In addition, cancer subtypes were infrequent and unevenly distributed across clusters, with several categories containing very small or zero counts. Under these conditions, formal inferential testing for cancer outcomes was not statistically appropriate. Therefore, cancer prevalence was presented descriptively and used solely for external validation of cluster-level patterns, rather than for hypothesis testing or estimation of effect sizes.

Response to E-4 (Post-hoc testing)

The Kruskal–Wallis test was applied as a global screening procedure to assess whether overall distributional differences existed among clusters. Because the primary aim of this study was exploratory characterization of cluster-level patterns rather than identification of specific pairwise group differences, post-hoc testing was not performed. **Accordingly, in the Results section, we avoided pairwise comparisons and restricted statistical reporting to overall group-level differences, without highlighting specific between-cluster contrasts.** This analytical choice is now explicitly stated in the Methods (Statistical Processing and Software).

Response to E-5 (Qualitative comparisons and statistical reporting)

We revised the Results section to remove impressionistic or qualitative comparisons that were not directly supported by statistical summaries. Specifically, in the subsections “**Clinical and biochemical characteristics across clusters**” and “**Cluster-level comparison of biomarkers**”, descriptive statements based solely on visual inspection of group means were eliminated or rewritten to reference distributional statistics and results of global tests. As a result, all cross-cluster comparisons are now consistently framed in terms of overall distributional differences, rather than subjective or qualitative descriptions, in line with the exploratory and non-pairwise analytical framework of the study.

Response to E-6 (Definition of “complete data” and handling of missing values)

In response to the editorial and reviewer comments, we revisited the definition of “complete data” and the handling of missing values to adopt the most transparent and methodologically defensible criteria. Specifically, in the **Methods (Study population and data preprocessing)**, we clarified that “complete data” refers to the availability of age and BMI, which were required to define the final analytical cohort. As a result of this clarification, the cohort was fixed at **N = 5,854**, and this sample size was used consistently throughout all subsequent analyses.

Missing values in other laboratory variables were addressed only after cohort definition, during the preprocessing stage, and did not alter the number of included participants. Importantly, under this stricter and more explicit data-handling framework, the clustering structure and the key finding of a distinct low-testosterone cluster characterized by differences in renal function remained robust, supporting the appropriateness of the revised approach.

Response to E-7 (Extent of missingness, justification of mean imputation, and impact on clustering)

We explicitly reported the extent of missingness for each variable in the **Methods (Study population and data preprocessing)** to improve

transparency. Mean imputation was selected as a conservative and assumption-minimizing approach to preserve the full analytical cohort after cohort definition, rather than to optimize predictive performance. Importantly, we verified that this imputation strategy did not materially alter the clustering structure. Under the revised and more explicit handling of missing data, the overall cluster configuration—including the separation of a low-testosterone group characterized by differences in renal function—remained stable, supporting the robustness of the findings to the chosen imputation method.

Response to E-8 (Placement of missing data handling in the Methods)

In line with the editorial and reviewer comments requesting earlier clarification, we moved the description of missing data handling to the initial part of the Methods section (**Step 1: Study population and data preprocessing**). By presenting the cohort definition, extent of missingness, and imputation strategy at this early stage, the analytical flow—from cohort selection to clustering—can now be followed more transparently, without ambiguity regarding sample size or preprocessing decisions.

Response to E-9 (Inconsistencies in cancer prevalence data)

The inconsistencies in cancer prevalence values identified by the reviewers were due to typographical errors in the text. To address this, we rechecked the original source data and verified that the values reported in **Table 2** accurately reproduce the underlying lifetime cancer prevalence data, including rounding. After confirmation, the corresponding text was corrected to ensure full consistency with the table. This revalidation confirms that the lifetime cancer data used for external validation are accurate and internally consistent.

Response to E-10 (Table consolidation and simplification of cancer outcomes)

To improve clarity and readability, we consolidated the previously separate cancer tables into a single, simplified table focusing on major and clinically common cancer categories. This revision was motivated by the primary goal of this analysis: to present external validation of the biomarker-defined low-testosterone, kidney-impaired cluster in a clear and accessible manner, rather than to exhaustively enumerate sparse cancer subtypes. Lifetime and recent cancer prevalence are now presented together to facilitate direct comparison across groups, and rare or heterogeneous cancer categories were grouped as "Other." As a result, the external validation findings can be more readily interpreted by readers, while preserving consistency with the underlying data.

Response to E-11 (Figures improved for clarity)

Thank you for this comment. We revised the figures and their presentation to improve readability and to ensure that each figure communicates a single, unambiguous message aligned with the new three-step structure (Step 1–3). Specifically, we (i) clarified the role of PCA as *visualization only* and corrected the variance explanation in the main text and Figure 3 caption (PC1/PC2 and cumulative variance are now explicitly reported), (ii) strengthened the justification of K=4 by presenting elbow and silhouette together as Figure 2 with consistent interpretation in the corresponding Results subsection ("Clustering of Male Subjects Using K-means"), and (iii) revised the network figure (Figure 4) and its caption to reduce ambiguity and improve interpretability by clearly defining nodes/edges and emphasizing its exploratory purpose. Overall, the revised figures are now more consistent with the manuscript's analytical flow and are easier to follow than in the previous version.

Response to E-12 (Reporting direction and magnitude of correlations)

To improve interpretability, we added a brief numerical reference to the Results to indicate the direction and approximate magnitude of representative correlations. Specifically, in **Results, Step 2 (Graph Theory–Based Correlation Structure Analysis)**, we now report example Spearman correlation coefficients

between total testosterone and liver enzymes (AST and ALT) in Cluster 0. These variables were selected as representative, non-definitional examples that illustrate the correlation structure without implying causality or shifting the focus toward primary outcomes. Full cluster-specific correlation coefficients are provided in Supplementary Table S3. Correlations involving key defining variables such as creatinine or CRP were intentionally not emphasized numerically in the main text to avoid overinterpretation beyond the exploratory scope of the analysis.

Response to E-13 (Clinical thresholds and translational relevance)

We appreciate the editor's suggestion regarding clinical relevance. The thresholds applied in this study (total testosterone <300 ng/dL, CRP >3 mg/L, and serum creatinine >1.2 mg/dL) were not derived from the present data, but correspond to commonly used clinical reference values. These cutoffs were employed solely to define biomarker-based profiles for descriptive characterization and external validation, rather than to propose new screening criteria or clinical decision thresholds. Given the cross-sectional and exploratory nature of the study, we intentionally avoided introducing additional clinical recommendations in the main text.

Response to E-14 (Cautious interpretation of cross-sectional findings)

We clarified that cancer outcomes were not selected as primary endpoints but were used solely for external validation of biomarker-defined cluster patterns. Cancer was chosen because it represents a clinically meaningful and well-documented outcome available in the external dataset, allowing assessment of whether the identified high-risk biomarker profile shows external consistency. This rationale is now explicitly stated in the Results (Step 3) and Discussion. To further improve clarity and context, we incorporated representative supporting references in the Discussion (e.g., Coussens & Werb, 2002; Yeap, 2009; Grandys et al., 2021; Yarmolinsky et al., 2024; Mok et al., 2025).

Response to E-16 (Linking biomarker patterns to biological mechanisms)

To strengthen biological context without overstating mechanistic claims, we added a concise discussion linking low testosterone, chronic inflammation, and impaired renal function to disease risk based on established literature. These associations are presented as contextual background rather than mechanistic proof, and we explicitly state that causal mechanisms cannot be inferred from the present cross-sectional data. To support this context, we incorporated representative references in the Discussion addressing testosterone deficiency and inflammation (e.g., Yeap, 2009; Grandys et al., 2021), as well as links between chronic inflammation, renal dysfunction, and cancer risk (e.g., Coussens & Werb, 2002; Mok et al., 2025).

Response to E-17 (Streamlining redundant methodological descriptions)

Redundant methodological explanations were streamlined throughout the manuscript. Standard or widely established procedures were condensed, while analytically critical components—such as the rationale for selecting $K=4$ clusters and the positioning of PCA as an exploratory visualization tool—were expanded and clarified. In addition, the entire manuscript was reorganized into a unified three-step framework (Step 1: unsupervised clustering, Step 2: cluster-specific network characterization, Step 3: external validation), allowing readers to follow the analytical logic in a more structured and rational manner. This restructuring improves readability and focuses attention on methodological decisions central to the study.

Response to E-18 (Replacing vague language with precise statistical reporting)

In response to this comment, we systematically revised the manuscript to replace vague or impressionistic language with precise statistical descriptions. Descriptive statements are now explicitly linked to predefined analyses, and interpretations are restricted to results supported by the stated statistical framework. In particular, qualitative expressions based solely on visual inspection (e.g., “appeared higher” or “seemed different”) were removed or reformulated to reflect overall distributional differences identified through global tests, or were clearly labeled as exploratory observations. These revisions improve statistical clarity and prevent overinterpretation.

Response to E-19 (Clarification of PCA variance explanations)

We clarified the role and limitations of PCA in both the Methods and Results. The revised manuscript explicitly reports the variance explained by PC1 and PC2 and clearly states that these components capture only a limited proportion of total variance. Accordingly, PCA is positioned solely as a tool for visualization and exploratory inspection, while cluster assignment and interpretation are based on analyses conducted in the full standardized feature space. Following revisions to the handling of missing data, PCA and the corresponding figures were recalculated; however, the overall structure and interpretative meaning of the PCA visualization remained unchanged, supporting the robustness of the presentation.

Response to E-20 (Removal of unnecessary terms such as

“doctors”)

As suggested, we removed unnecessary or potentially restrictive terms such as “doctors” from the manuscript, particularly in the Plain Language Summary. The revised wording avoids implying a narrow professional audience and more accurately reflects the broader relevance of the findings to researchers, clinicians, and other stakeholders.

Response to E-21 (Consistency in terminology and formatting across the manuscript)

We conducted a comprehensive review of the entire manuscript to ensure consistency in terminology, abbreviations, and formatting across all sections, tables, and figures. Cluster labels, statistical terms, and variable names were standardized, and section headings and figure legends were harmonized to improve clarity and coherence. These revisions ensure that the manuscript presents a consistent and unified structure throughout.

Editorial Summary of Revisions (for the Reviewer #1)

We sincerely thank Reviewer #1 for the careful and insightful evaluation of our manuscript. The reviewer raised several important points that helped us substantially improve both the clarity and the methodological positioning of the study. In response, we addressed these comments along three main lines.

First, we clarified the role of dimensionality reduction and clustering by explicitly distinguishing between exploratory visualization (PCA) and formal cluster assignment performed in the full multivariate space. This clarification was incorporated throughout the Methods and Results to prevent overinterpretation of low-dimensional projections.

Second, we strengthened the statistical framing of the analysis by clearly defining the exploratory nature of the study. We revised the presentation of statistical comparisons to avoid impressionistic or qualitative interpretations, restricted inference to global distributional differences, and clarified the scope and limitations of the applied statistical tests.

Third, we revised the overall structure and presentation of the manuscript to improve readability and coherence. This included reorganizing the Results into a stepwise analytical framework, refining figure and table presentation, and adopting more cautious language in the interpretation of cross-sectional findings.

We believe that these revisions have significantly improved the transparency, rigor, and interpretability of the manuscript, and we are grateful to the reviewer for comments that directly contributed to a clearer and more robust presentation of our work.

Response to R1-1 (Limited cluster separation in PCA visualization)

We agree that the PCA plot does not show clear visual separation between clusters. This overlap is expected because the first two principal components explain only a limited proportion of the total variance, and PCA was not used for cluster determination. We therefore clarified in the Methods and Results that PCA was applied solely for exploratory visualization, while cluster assignment was performed using K-means clustering in the full standardized multivariate space. Importantly, the absence of clear separation in the two-dimensional PCA projection reflects the high-dimensional nature of the data and does not contradict the presence of clinically meaningful cluster structure identified in the full feature space.

Response to R1-2 (Limited variance explained by PCA)

We agree that the first two principal components explain only a limited proportion of the total variance (~33%). To address this point, we revised the manuscript to explicitly state the proportion of variance explained and to clarify the intended role of PCA as a tool for visualization and exploratory inspection only, rather than for supporting cluster determination. This clarification helps readers understand why limited separation in the PCA projection is expected and directs interpretation toward clustering results obtained in the full standardized feature space, thereby improving transparency and preventing overinterpretation of low-dimensional representations.

Response to R1-3 (Small size of Cluster 3 and its robustness)

We agree that the very small size of Cluster 3 raises concerns regarding its statistical and biological robustness. In the revised manuscript, we explicitly clarified that this cluster represents a singleton subgroup and is not used for inferential statistical comparisons. Instead, it is retained solely as an exploratory observation to illustrate potential heterogeneity within the dataset. To avoid overinterpretation, this cluster is clearly distinguished from the main clusters throughout the Results and excluded from comparative statistical analyses, ensuring that the primary conclusions of the study are driven by well-populated and robust clusters.

In addition, this singleton cluster emerged after reanalysis conducted in response to editorial recommendations regarding the handling of missing data. Under the revised and more stringent preprocessing criteria, a small number of cases with incomplete key variables were no longer eligible for inclusion. Importantly, the primary cluster structure—particularly the group characterized by low testosterone and impaired renal function—remained clearly identifiable. The reduction of this minor subgroup to a single case thus reflects increased analytical rigor following editorial guidance, rather than a change in the main findings, which were robust to this reanalysis.

Response to R1-4 (Interpretation of Kruskal–Wallis test results)

The Kruskal–Wallis test evaluates whether at least one distribution differs among clusters, but it does not identify which specific clusters differ from each other. In this study, the test was positioned as a **global screening procedure** to assess the presence of overall distributional differences across clusters, rather than to detect specific pairwise contrasts. Accordingly, interpretation in the Results is restricted to overall distributional patterns at the cluster level, without implying individual cluster-to-cluster differences.

Response to R1-5 (Qualitative interpretation without formal statistical support)

We agree that portions of the original Discussion relied on qualitative descriptions that were not sufficiently anchored to formal statistical analysis. In the revised manuscript, we restructured the Results and Discussion to ensure that all descriptive comparisons are explicitly grounded in predefined statistical analyses or clearly labeled as exploratory observations. Qualitative language based solely on visual inspection of group means was removed or replaced with statistically appropriate descriptions, and interpretations are now restricted to overall distributional differences identified through the Kruskal–Wallis test. This revision improves rigor and prevents overinterpretation of descriptive trends.

Response to R1-6 (Cross-sectional design and precision medicine)

We agree with the reviewer that the cross-sectional and observational nature of this study precludes causal inference and limits direct clinical actionability. In the revised manuscript, this limitation is explicitly acknowledged in both the Introduction and Discussion. Importantly, we confirmed that the manuscript does not frame the findings in terms of individualized treatment or actionable precision medicine. Instead, the study is positioned as a descriptive, data-driven characterization of physiological heterogeneity and biomarker-defined risk profiles. References to “precision” are used strictly in the context of phenotypic stratification rather than causal or interventional claims, and the need for longitudinal or interventional studies to establish causality is clearly stated. We therefore believe that the current wording appropriately reflects the scope and limitations of the study design.

Editorial Summary of Revisions (for the Reviewer #2)

We sincerely thank Reviewer #2 for the thoughtful and constructive evaluation of our manuscript, as well as for the balanced assessment of its strengths and limitations. The reviewer's comments were particularly helpful in refining both the methodological clarity and overall presentation of the study. In response, we addressed these comments along three main directions.

First, we clarified the rationale for selecting cancer as the outcome for external validation and strengthened the explanation of how this choice relates to the study objective of assessing the consistency and relevance of biomarker-defined clusters across populations.

Second, we improved methodological transparency by clarifying the definition of "complete data," reporting missingness, and explicitly describing the timing and rationale of missing data imputation within the analytical workflow.

Third, we reorganized the manuscript into a unified three-step framework (Step 1: unsupervised clustering, Step 2: cluster-specific network characterization, Step 3: external validation). This structure was applied consistently across the Introduction, Methods, Results, and Discussion, with revised subsection titles, allowing readers to more easily follow the overall analytical logic and flow of the study.

We believe that these revisions have substantially improved the clarity, coherence, and accessibility of the manuscript, and we are grateful to the reviewer for comments that directly contributed to strengthening the work.

Response to R2-1 (Rationale for selecting cancer as the outcome in external validation)

We appreciate the reviewer's request to clarify the rationale for selecting cancer as the outcome for external validation. In the revised manuscript, we explicitly explain that cancer was not chosen as a primary endpoint, but as a clinically

meaningful and well-documented outcome available in the NHANES dataset, suitable for assessing the external consistency of biomarker-defined clusters.

Importantly, this choice is supported by prior literature demonstrating established associations between testosterone deficiency, systemic inflammation, renal dysfunction, and cancer risk. Previous studies have reported links between low testosterone and multi-organ or inflammatory dysregulation (e.g., Yeap, 2009; Grandys et al., 2021; Rotter et al., 2025), between chronic inflammation and carcinogenesis (e.g., Coussens & Werb, 2002; Yarmolinsky et al., 2024), and between impaired kidney function and increased cancer incidence (e.g., Mok et al., 2025). Together, these findings provide a biological and epidemiological context supporting the use of cancer prevalence as an external reference outcome, without implying causality.

Accordingly, cancer outcomes were used solely in a descriptive manner to evaluate whether the high-risk biomarker profile identified in the Japanese cohort—particularly the combination of low testosterone, elevated inflammation, and impaired renal function—showed consistent associations with disease burden in an independent population. This rationale is now clarified in both the Introduction and the External Validation section of the Results.

Response to R2-2 (Clarification of “complete data” and missing value imputation)

We appreciate the reviewer’s careful reading and agree that the description of “complete data” and subsequent missing value imputation required clarification. In the revised manuscript, we explicitly distinguish between variables used for cohort inclusion and variables included in downstream analyses. Specifically, “complete data” refers to the availability of age and BMI, which were required as essential baseline variables for inclusion in the analytical cohort. After applying this inclusion criterion, missing values in the remaining clinical and biochemical variables were handled by column-wise mean imputation to allow exploratory multivariate analyses without unnecessary loss of sample size.

Following editorial recommendations, we re-evaluated the handling of missing data and revised the Methods to describe the timing, rationale, and impact of imputation at an earlier stage. We also clarified that the primary cluster structure—particularly the cluster characterized by low testosterone and impaired renal function—remained robust under this revised preprocessing strategy. These revisions were implemented to improve transparency and avoid potential misinterpretation of the analytical workflow.

Response to R2-3 (Inconsistencies in cancer prevalence data)

We thank the reviewer for identifying inconsistencies between the cancer prevalence values reported in the text and those presented in Tables 2a and 2b. Upon careful re-examination, we confirmed that these discrepancies were due to typographical errors in the manuscript text rather than errors in the underlying data.

We re-verified all cancer prevalence values directly against the original NHANES dataset and confirmed that the values reported in Table 2 accurately reproduce the underlying data, including rounding to the reported decimal places. The text has been corrected accordingly to ensure full consistency with the tables. No changes were made to the data or analytical procedures, and the correction does not affect the interpretation or conclusions of the study.

Response to R2-4 (Mechanistic links between biomarkers and cancer outcomes)

We appreciate the reviewer's suggestion to strengthen the biological context linking the observed biomarker patterns to cancer outcomes. In response, we added a brief, focused discussion outlining plausible mechanistic pathways supported by existing literature, particularly those involving chronic inflammation, endocrine dysregulation, and impaired renal function. Importantly, this discussion is framed as contextual background rather than mechanistic proof, and no causal interpretation is implied.

Specifically, we note that testosterone deficiency has been associated with systemic inflammation and metabolic dysregulation, while chronic inflammation and reduced kidney function have independently been linked to increased cancer risk. These established associations provide a biologically plausible context for the observed cluster-level patterns in external validation. At the same time, we explicitly state that the present study is cross-sectional and descriptive, and that elucidation of causal mechanisms will require longitudinal or experimental studies. This clarification has been incorporated into the Discussion.

To support this contextual discussion, we added representative citations addressing testosterone–inflammation/metabolic dysregulation and inflammation/renal dysfunction in relation to cancer risk (e.g., Yeap, 2009; Grandys et al., 2021; Coussens & Werb, 2002; Yarmolinsky et al., 2024; Mok et al., 2025).

Response to R2-5 (Supplementary glossary)

We thank the reviewer for the positive comment regarding the supplementary glossary. We have retained the glossary as it is, as we agree that it improves accessibility for readers less familiar with the statistical techniques used in this study.

Response to R2-6 (Use of the term “doctors” in the Plain

Language Summary)

We appreciate the reviewer’s suggestion regarding wording in the Plain Language Summary. In the revised manuscript, we removed the term “doctors” from the final sentence to avoid implying a narrow professional audience. The wording was revised to reflect the broader relevance of the findings to researchers, clinicians, and other stakeholders.

Response to R2-7 (Clarification of PCA variance explanation)

We thank the reviewer for pointing out this inconsistency. In the revised manuscript, we removed the misleading expression that PCA1 and PCA2 accounted for the “majority” of the variance and instead explicitly report the variance explained by each component (PC1 and PC2, cumulative ~30%). We further clarified in both the Methods and Results that PCA was used exclusively for visualization and exploratory inspection, and that cluster assignment and interpretation were based on analyses conducted in the full standardized feature space. This revision ensures internal consistency and prevents overinterpretation of low-dimensional projections.

Response to R2-8 (Direction and magnitude of correlations between TT and liver markers)

We agree that indicating the direction and magnitude of representative correlations improves interpretability. In response, we added a brief statement in the Results (Step 2) noting that, in Cluster 0, total testosterone showed inverse correlations with liver-related enzymes (e.g., AST and ALT), and we provided representative r -values to indicate both direction and strength. To avoid overemphasis, these values are presented as illustrative examples, while full correlation coefficients for all variables and clusters are reported in Supplementary Table S3. This addition improves clarity while maintaining the exploratory scope of the analysis.

Editorial Summary of Revisions (for the Reviewer #3)

We sincerely thank Reviewer #3 for the constructive and helpful comments. In response, we revised the manuscript to improve clarity, conciseness, and methodological transparency, while carefully preserving the exploratory scope of the study.

A key revision was the reorganization of the manuscript into a clearly defined three-step analytical framework (Step 1: unsupervised clustering, Step 2: cluster-specific correlation and network characterization, and Step 3: external validation). This stepwise structure was applied consistently across the Introduction, Methods, Results, and Discussion, allowing the overall logic and analytical flow to be presented in an integrated and coherent manner. As a result, the roles of clustering, network analysis, and external validation are now more clearly distinguished, and their interrelationships are easier for the reader to follow.

Within this revised structure, redundancy in methodological explanations was reduced by consolidating overlapping descriptions, and the rationale for selecting $K = 4$ clusters was clarified by explicitly comparing $K = 3$ and $K = 4$ solutions and emphasizing clinical separability rather than optimization of a single clustering metric. The description of missing data handling was moved to an earlier position in the Methods to clearly present the analytical workflow and data preprocessing steps.

We also revised Figure 4 and its accompanying text to clearly position the network visualization as a qualitative representation of correlation structure rather than a basis for mechanistic interpretation. Issues related to multiple testing were addressed by explicitly framing correlation analyses as exploratory and limiting interpretation to overall structural patterns. In addition, language describing the identified clusters was revised to avoid disease labeling or causal implication, clarifying that the findings represent descriptive biomarker configurations rather than definitive aging phenotypes. Redundant tables were consolidated, and statements regarding clinical application were refined to emphasize that translational implications should be addressed in future longitudinal studies.

We appreciate Reviewer #3's thoughtful feedback, which helped substantially improve the coherence, rigor, and interpretability of the manuscript.

Response to R3-1 (Manuscript length and redundancy)

We agree that portions of the manuscript contained redundant methodological explanations. In the revised version, we streamlined the Methods and Discussion sections by removing overlapping descriptions and consolidating general methodological statements where appropriate. These changes improved overall clarity and conciseness without altering the analytical framework or interpretation.

At the same time, conceptually critical elements of the analysis were reorganized rather than simply shortened. In particular, key methodological decisions specific to this study—such as the rationale for selecting $K = 4$ clusters—were separated into dedicated paragraphs and explained in greater detail. This selective condensation of general descriptions, combined with focused elaboration of study-defining methodological choices, allowed the overall presentation to be more coherent and logically structured.

Response to R3-2 (Justification for $K = 4$ clusters)

We thank the reviewer for raising this important point. In the revised manuscript, we clarified that the selection of $K = 4$ was not based on a subjective preference but on an explicit evaluation of trade-offs among multiple, complementary criteria.

As described in the Methods, cluster number selection integrated (i) within-cluster sum of squares (elbow behavior), (ii) silhouette coefficients, (iii) stability across random initializations, and (iv) separability of biomarker-defined centroids. Although the average silhouette coefficient was marginally higher at $K = 3$, the absolute difference between $K = 3$ and $K = 4$ was small. In contrast, centroid-level comparison demonstrated that $K = 4$ yielded a reproducible

subdivision of individuals with low testosterone and elevated CRP into two subgroups that differed with respect to renal function, a distinction that was not captured in the $K = 3$ solution.

Importantly, following editorial guidance, missing data handling was revised and all clustering analyses were recalculated using the updated dataset. Under this more explicit and transparent preprocessing framework, the same biologically interpretable structure—namely, the separation of a low-testosterone group according to renal function—remained identifiable only in the $K = 4$ solution. This consistency supports the robustness of the $K = 4$ configuration and indicates that the result does not depend on a specific preprocessing choice.

Because the primary aim of the study was to characterize physiologically interpretable heterogeneity rather than to optimize a single clustering index, $K = 4$ was therefore adopted as the primary solution, with the smallest cluster retained solely as an exploratory subgroup and excluded from inferential comparisons.

Specifically, we introduced a new subsection in the Methods (“K-means clustering and justification for selecting $K = 4$ ”), in which the comparison between $K = 3$ and $K = 4$, the evaluation of trade-offs among clustering criteria, and the robustness of the resulting structure after revised missing-data handling are described in a standalone paragraph.

Response to R3-3 (Placement of missing data imputation)

We agree that transparency regarding data handling is essential. Accordingly, we moved the description of missing data handling to the beginning of the Methods section. This revision clarifies the analytical workflow by explicitly stating the inclusion criteria (complete age and BMI), the extent of remaining missingness, and the imputation strategy prior to clustering and downstream analyses, allowing readers to understand the full preprocessing framework before any multivariate analyses are introduced.

Response to R3-4 (Clarity of Figure 4)

We revised the description of Figure 4 to emphasize its role as a qualitative visualization of correlation structure rather than a basis for mechanistic interpretation. The revised text explicitly limits interpretation to structural patterns of association and avoids inference based on visual prominence, node positioning, or apparent network centrality.

In addition, the figure itself was simplified in the revised version by reducing visual complexity and presenting cluster-specific networks in separate panels, thereby improving readability while minimizing the risk of overinterpretation. These changes were made to reduce interpretive ambiguity and to align Figure 4 strictly with its exploratory purpose.

Response to R3-5 (multiple testing)

In the revised manuscript, correlation and network analyses are explicitly positioned as exploratory, with interpretation restricted to overall structural patterns rather than individual pairwise associations. To reduce noise and spurious connections, a correlation threshold was applied before network construction.

Because the purpose of these analyses was descriptive characterization rather than hypothesis testing, formal multiple testing correction was not applied by design. This rationale is now clearly stated in the Methods and Results to avoid overinterpretation of individual correlations.

Response to R3-6 (Interpretation of “composite aging”)

In the original version, we used the term “composite aging” and described Cluster 0 in a way that could be interpreted as a discrete phenotype or disease-like construct. In the revised manuscript, we removed this labeling and revised the Discussion to explicitly frame Cluster 0 as a descriptive configuration of biomarker distributions rather than a disease entity or a definitive aging phenotype, emphasizing the cross-sectional and hypothesis-generating nature of the findings. We also clarified that network centrality reflects statistical association structure and should not be interpreted as biological hierarchy or

causal influence (e.g., von Elm et al., 2008; Dablander and Hinne, 2019; Borsboom et al., 2021).

Response to R3-7 (Redundant tables)

In the original version, several tables contained overlapping information and parallel presentations of related variables, which reduced readability. In the revised manuscript, we consolidated these tables while retaining all clinically relevant information.

Specifically, the cluster characteristics table was reconstructed after reanalysis based on revised missing-data handling, resulting in a single, streamlined Table 1 that reports key demographic, anthropometric, biochemical, inflammatory, renal, and hormonal variables using median and interquartile ranges, with the singleton cluster excluded from inferential comparisons. This revision replaces multiple partially redundant summaries with a single coherent presentation aligned with the final analytical cohort.

Similarly, for the external validation, two separate cancer tables with extensive site-specific listings were merged into a single table. Cancer categories were consolidated into clinically meaningful groups to highlight major site-specific patterns while avoiding unnecessary fragmentation. Both lifetime cancer prevalence and recent cancer diagnoses are now presented within a unified structure, improving clarity and interpretability without loss of information.

Overall, these changes reduced redundancy, corrected inconsistencies, and improved the clarity of data presentation while preserving all essential results.

Response to R3-8 (Clinical applications)

In the revised manuscript, we explicitly position the study as descriptive and exploratory. Rather than proposing specific biomarker thresholds or clinical interventions, we state that potential clinical applications should be addressed in future longitudinal or interventional studies. In the Discussion, we therefore emphasize future directions—such as network-informed multivariate risk stratification and validation using repeated measurements over time—while

avoiding premature translation from cross-sectional findings (e.g., López-Otín et al., 2023; Ferrucci et al., 2021; Barabási et al., 2011).